# Ease-of-Teaching and Language Structure from Emergent Communication

**Fushan Li**
Department of Computing Science
University of Alberta
Edmonton, Canada
fushan@ualberta.ca

**Michael Bowling**
Department of Computing Science
University of Alberta
Edmonton, Canada
mbowling@ualberta.ca

## Abstract

Artificial agents have been shown to learn to communicate when needed to complete a cooperative task. Some level of language structure (e.g., compositionality) has been found in the learned communication protocols. This observed structure is often the result of specific environmental pressures during training. By introducing new agents periodically to replace old ones, sequentially and within a population, we explore such a new pressure — ease of teaching — and show its impact on the structure of the resulting language.

## 1 Introduction

Communication among agents is often necessary in partially observable multi-agent cooperative tasks. Recently, communication protocols have been automatically learned in pure communication scenarios such as referential games (Lazaridou et al., 2016; Havrylov and Titov, 2017; Evtimova et al., 2018; Lee et al., 2018) as well as alongside behaviour polices (Sukhbaatar et al., 2016; Foerster et al., 2016; Lowe et al., 2017; Grover et al., 2018; Jaques et al., 2019).

In addition to demonstrating successful communication in different scenarios, one of the language properties, compositionality, is often studied in the structure of the resulting communication protocols (Smith et al., 2003a; Andreas and Klein, 2017; Bogin et al., 2018). Many works (Kirby et al., 2015; Mordatch and Abbeel, 2018) have illustrated that compositionality can be encouraged by environmental pressures. Under a series of specific environmental pressures, Kottur et al. (2017) have coaxed agents to communicate the compositional atoms each turn independently in a grounded multi-turn dialog. More general impact of environmental pressures on compositionality is investigated in Lazaridou et al. (2018) and Choi et al. (2018), including different vocabulary sizes, different message lengths and carefully constructed distractors.

Compositionality enables languages to represent complex meanings using meanings of its components. It preserves the expressivity and compressibility of the language at the same time (Kirby et al., 2015). Moreover, often emergent languages can be hard to interpret for humans. With the compressibility of fewer rules underlying a language, emergent languages can be easier to understand. Intuitively, a language that is more easily understood should be easier-to-teach as well. We explore the connection between ease-of-teaching and the structure of the language through empirical experiments. We show that a compositional language is, in fact, easier to teach than a less structured one.

To facilitate the emergence of easier-to-teach languages, we create a new environmental pressure for the speaking agent, by periodically forcing it to interact with new listeners during training. Since the speaking agent needs to communicate with new listeners over and over again, it has an encouragement to create a language that is easier-to-teach. We explore this idea, introducing new listeners periodically to replace old ones, and measure the impact of the new training regime on ease-of-teaching of the

language and its structure. We show that our proposed reset training regime not only results in easier-to-teach languages, but also that the resulting languages become more structured over time.

A similar regime, iterated learning (Smith et al., 2003b), has been studied in the language evolution literature for decades (Kirby, 1998; Smith and Hurford, 2003; Vogt, 2005; Kirby et al., 2008, 2014), as an experimental paradigm for studying the origins of structure in human language. Their works involve the iterated learning of artificial languages by both computational models and humans, where the output of an individual is the input for another. The main difference from our work is that we have only one speaker and languages evolve by speaking to new listeners, while iterated learning emphasizes language transmission and acquisition from speaker to speaker. We also focus on deep neural network architectures trained by reinforcement learning methods.

Our experiments sequentially introducing new agents suggest that the key environmental pressure — ease of teaching — may come from learning with a population of other agents (e.g., Jaderberg et al., 2019). Furthermore, an explicit (and large) population may be beneficial to smooth out abrupt changes to the training objective when new learners are added to the population. However, in a second set of experiments we show that these "advantages" surprisingly actually remove the pressure to the speaker. In fact, just the opposite happens: more abrupt changes appear to be key in increasing the entropy of the speaker's policy, which seems to be key to increasingly structured, easier-to-teach languages.

## 2 Experimental Framework

We explore emergent communication in the context of a referential game, which is a multi-agent cooperative game that requires communication between a speaker $S$ and a listener $L$. We first describe the setup of the game and then the agent architectures, training procedure, and evaluation metrics we use in this work.

### 2.1 Game Setup

The rules of the game are as follows. The speaker is presented with a target object $t \in O$ and sends a message $m$ to a listener using a fixed-length ($l = 2$) sequence of symbols $(m_1, m_2)$ from a fixed-sized vocabulary ($m_i \in V$ where $|V| = 8$). The listener is shown candidate objects $C = \{c_1, \ldots, c_5\} \subseteq O$ where $t \in C$ and 4 randomly selected distracting objects, and must guess $\hat{t}$ which is the target object. If the listener guesses correctly $\hat{t} = t$, the players succeed and get rewarded $r = 1$, otherwise, they fail and get $r = 0$.

Each object in our game has two distinct attributes, which for ease of presentation we refer to as color and shape. There are 8 colors (i.e., black, blue, green, grey, pink, purple, red, yellow) and 4 shapes (i.e., circle, square, star, triangle) in our setting, therefore $8 \times 4 = 32$ possible different objects. For simplicity, each object is represented by a 12-dimension vector concatenating a one-hot vector of color with a one-hot vector of shape, which is similar to Kottur et al. (2017).[1]

### 2.2 Agent Architecture

We model the agents' communication policies $\pi^S$ and $\pi^L$ with neural networks similar to Lazaridou et al. (2018) and Havrylov and Titov (2017). For the speaker, $\pi^S$ stochastically outputs a message $m$ given $t$. For the listener, $\pi^L$ stochastically selects $\hat{t}$ given $m$ and all the candidates $C$. In the following, we use $\theta^S$ to represent all the parameters in the speaker's policy $\pi^S$ and $\theta^L$ for the listener's policy $\pi^L$. The architectures of the agents are shown in Figure 1.

Concretely, for the speaker, $f^S(t; \theta^S)$ obtains an embedding $u_t$ of the target object $t$. At the first time step $\tau = 0$ of LSTM (Hochreiter and Schmidhuber, 1997) $g^S$, we initialize $u_t$ as the start hidden state $h_0^S$ and feed a start token $\langle S \rangle$ (viz., a zero vector) as the input of $g^S$. At the next step $\tau + 1$, $o^S(h_{\tau+1}^S; \theta^S)$ performs a linear transformation from $h_{\tau+1}^S$ to the vocabulary space, and then applies a softmax function to get the probability distribution of uttering each symbol in the vocabulary. The next token $m_{\tau+1}$ is sampled from the probability distribution over the vocabulary and serves as additional input to $g^S$ at the next time step $\tau + 1$ until a fixed message length $l$ is reached. For

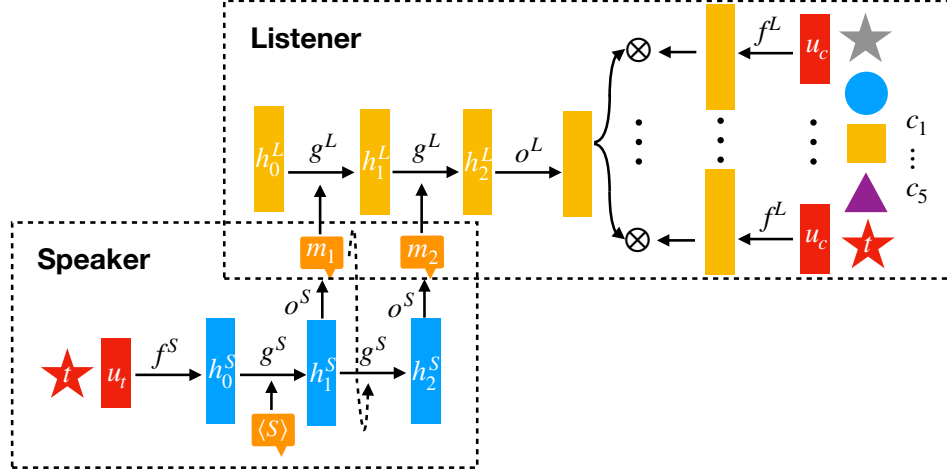

Figure 1: The architectures of the agents.

the listener, the input to the LSTM $g^L(m, h^L; \theta^L)$ are tokens received from the speaker and all the candidate objects are represented as embeddings $u_c$ using $f^L(c; \theta^L)$. $o^L(h^L; \theta^L)$ transforms the last hidden state of $g^L$ to an embedding and a dot product with $u_{c_1}, \ldots, u_{c_5}$ is performed. We then apply a softmax function to the dot products to get the probability distribution for the predicted target $\hat{t}$. During evaluation, we use argmax instead of softmax for both $S$ and $L$, which results in a deterministic language. The dimensionalities of the hidden states in both $g^S$ and $g^L$ are 100.

## 2.3 Training

In all of our experiments, we use stochastic gradient descent to train the agents. Our objective is to maximize the expected reward under the agents' policies $J(\theta^S, \theta^L) = \mathbb{E}_{\pi^S, \pi^L}[R(\hat{t}, t)]$. We compute the gradients of the objective by REINFORCE (Williams, 1992) and add an entropy regularization (Mnih et al., 2016) term[2] to the objective to maintain exploration in the policies:

$$\nabla_{\theta^S} J = \mathbb{E}_{\pi^S, \pi^L}[R(\hat{t}, t) \cdot \nabla_{\theta^S} \log \pi^S(m|t)] + \lambda^S \cdot \nabla_{\theta^S} H[\pi^S(m|t)]$$
$$\nabla_{\theta^L} J = \mathbb{E}_{\pi^S, \pi^L}[R(\hat{t}, t) \cdot \nabla_{\theta^L} \log \pi^L(\hat{t}|m, c)] + \lambda^L \cdot \nabla_{\theta^L} H[\pi^L(\hat{t}|m, c)]$$

where $\lambda^S, \lambda^L > 0$ are hyper-parameters and $H$ is the entropy function.

For training, we use the Adam (Kingma and Ba, 2014) optimizer with learning rate 0.001 for both $S$ and $L$. We use a batch size of 100 to compute policy gradients.

We use $\lambda^S = 0.1$ and $\lambda^L = 0.05$ in all our experiments. Discussion about how $\lambda^S$ and $\lambda^L$ were chosen is included in Appendix A. All the experiments are repeated 1000 times independently with the same random seeds for different regimes, unless we specifically point out the number of experimental trials. In all figures, the solid lines are the means and the shadings show a 95% confidence interval (i.e., 1.96 times the standard error), which in many experiments is too tight to observe.

## 2.4 Evaluation

We evaluated the emergent language in two ways, ease-of-teaching and the degree of compositionality of the language.[3]

To evaluate the ease-of-teaching of the resulting language (i.e., a new listener reaches higher task success rates with less training), we keep the speaker's parameters unchanged and produce a deter-

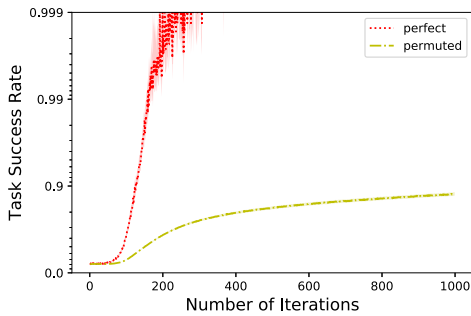

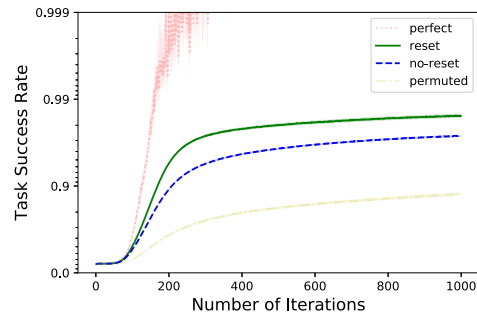

Figure 2: Ease-of-teaching of the artificial languages.

Figure 3: Ease-of-teaching of the emergent languages.

ministic language with argmax, and then train 30 new randomly initialized listeners with the speaker for 1000 iterations and observe how quickly the task success is achieved.

To the best of our knowledge, there is not a definitive quantitative measure of language compositionality. However, a quantitative measure of message structure, i.e., topographic similarity, exists in the language evolution literature. Seeing emergent languages as a mapping between the meaning space and the signal space, topographic similarity is defined as the correlation between distances of pairs in the meaning space and those in the signal space (Brighton and Kirby, 2006). We use topographic similarity to measure the structural properties of the emergent languages quantitatively as in Kirby et al. (2008) and Lazaridou et al. (2018). We compute this measure as follows: we exhaustively enumerate all target objects and the resulting messages from the deterministic $\pi^S$. We compute the cosine similarities $s$ between all pairs of objects' vector representations and the Levenshtein distances $d$ between all pairs of objects' messages. The topographic similarity is calculated as the negative Spearman $\rho$ correlation between $s$ and $d$. The higher the topographic similarity is, the higher the degree of compositionality in the language.

## 3 Compositionality and Ease-of-Teaching

In the introduction, we hypothesized that a compositional language is easier to teach than a less structured language. To test this, we construct two artificial languages, a perfect language with topographic similarity 1.0 and a permuted language. We create a perfect language by using 8 different symbols (a-h) from the vocabulary to describe 8 shapes, and choose 4 symbols (a-d) to represent 4 colors. The permuted language is formed by randomly permuting the mappings between messages and objects from the perfect language. It still represents all objects with distinctive messages, but each symbol in a message does not have a consistent meaning for shape and color. For example, 'aa' means 'red circle', 'ab' can mean 'black square'. We then teach both languages to 30 randomly initialized listeners and observe on average how fast the languages can be taught. A language that is easier-to-teach than another, means reaching higher task success rates with less training.

We generated 1 perfect language and 100 randomly permuted languages, which had an average topographic similarity of 0.13. The training curves for both languages are plotted in Figure 2. We can see that the listener learns a compositional language much faster than a less structured language.

## 4 Experiments with Listener Reset

After observing a compositional language is easier to teach than a less structured one, we now explore a particular environmental pressure for encouraging the learned language to favour one that is easier to teach. The basic idea is simple, forcing the speaker to teach its language over and over again to new listeners.

To facilitate the emergence of languages that are easier to teach, we design a new training regime: after training a speaker $S$ and a listener $L$ to communicate for a fixed number of iterations, we randomly reinitialize a new listener $L'$ to replace the old one and have the speaker $S$ and the new listener

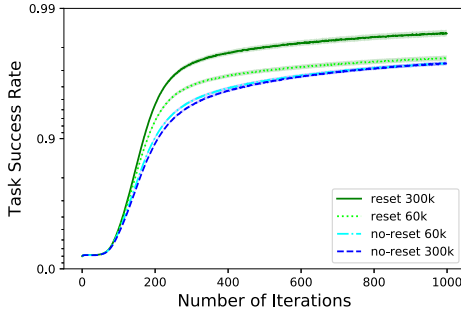

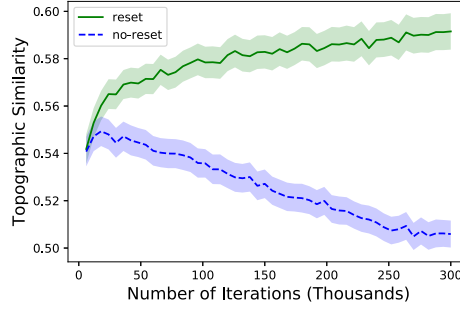

Figure 4: Ease-of-teaching of the emergent languages during training.

Figure 5: Comparison of topographic similarity under reset and no-reset regime.

$L'$ continue the training process. We repeat doing this replacement multiple times. We name this process "reset". Since the speaker needs to be able to communicate with a newly initialized listener periodically, this hopefully gives the speaker an environmental pressure to favour an easier-to-teach language.

We explore this idea by training a speaker with 50 listeners sequentially using the proposed reset regime and a baseline method with only one listener for the same number of iterations, which we call "no-reset". In the reset regime, the listener is trained with the speaker for 6k (i.e., 6000) iterations and then we reinitialize the listener. Thus, the total number of training iterations is $6k \times 50 = 300k$. In the no-reset regime, we train a speaker with the same listener for 300k iterations.

## 4.1 Ease-of-teaching under the Reset Training Regime

After training, both methods can achieve a communication success rate around 97% at evaluation. In fact, after around 6k iterations agents can achieve high communication success rate, but in this work we are interested in how language properties are affected by the training regime. Therefore, the following discussion is not about the communication success, but the differences between the emergent languages in terms of their ease of teaching and the degree of compositionality. We first evaluate the ease-of-teaching of the resulting language after 300k iterations of training and show the results in Figure 3. We can see that the languages emergent from the reset regime are on average easier to teach than without resets.

We also test the ease-of-teaching of the emergent languages every 60k training iterations, to see how ease-of-teaching changes during training. Results are shown in Figure 4, although for simplicity we are showing the ease of teaching for just one intermediate datapoint, viz., after 60k iterations. For the reset regime, the teaching speed of the language is increasing with training. For the no-reset regime, the teaching speed is in fact getting slower.

## 4.2 Structure of the Emergent Languages

But does the emergent language from the reset regime also have a higher degree of compositionality? We compute the topographic similarity of the emergent languages for both methods every 6k iterations (before the listener in the reset regime gets reset), and show how the topographic similarity evolves during training in Figure 5. In the reset regime, the topographic similarity rises with training to 0.59; while in the no-reset regime the metric drops with training to 0.51. This shows that not only are the languages getting easier to teach with additional resets, but the languages are getting more structured, although still on average far from a perfectly structured language.

Table 1 shows an example of one of the resulting languages from the reset regime. This is from an above average outcome where the topographic similarity is 0.97. In this example, each color is represented by a separate unique symbol as $m_1$. Each shape is represented by a disjoint set of symbols as $m_2$. {'c', 'a'} can represent 'circle' ('a' is used only once in 'yellow circle'), 'g' means 'square', 'd' for 'star' and 'b' for 'triangle'.

Table 1: A learned language with topographic similarity 0.97 from the reset regime

|          | black | blue | green | grey | pink | purple | red | yellow |
|----------|-------|------|-------|------|------|--------|-----|--------|
| circle   | bc    | hc   | dc    | ac   | fc   | ec     | cc  | ga     |
| square   | bg    | hg   | dg    | ag   | fg   | eg     | cg  | gg     |
| star     | bd    | hd   | dd    | ad   | fd   | ed     | cd  | gd     |
| triangle | bb    | hb   | db    | ab   | fb   | eb     | cb  | gb     |

Table 2: A learned language with topographic similarity 0.83 from the no-reset regime

|          | black | blue | green | grey | pink | purple | red | yellow |
|----------|-------|------|-------|------|------|--------|-----|--------|
| circle   | bg    | bf   | bh    | bd   | ba   | be     | bc  | bb     |
| square   | cg    | hf   | hh    | cd   | ca   | he     | hc  | cb     |
| star     | eg    | af   | eh    | ed   | da   | ae     | dc  | ab     |
| triangle | gg    | gf   | gh    | gd   | gh   | ge     | gc  | gb     |

The most structured language from the no-reset regime after 300k iterations, with topographic similarity 0.83, is shown in Table 2. In this language, the first letter $m_1$ means shape, 'b' for 'circle', {'c', 'h'} for 'square', {'e', 'a', 'd'} for 'star', 'g' for 'triangle'. As for the color $m_2$, messages are mostly separable except 'h' are shared between 'green triangle' and 'pink triangle', which makes the language ambiguous.

## 5 Experiments with a Population of Listeners

We have so far shown introducing new listeners periodically creates a pressure for ease-of-teaching and more structure. One might expect that learning within an explicit population of diverse listeners (e.g., each having experienced different number of training iterations) could increase this effect. Furthermore, one might expect a larger population could smooth out abrupt changes to the training objective when replacing a single listener.

### 5.1 Staggered Reset in a Population of Listeners

We explore this alternative in our population training regime. We now have $N$ listeners instead of 1, and each listener's lifetime is fixed to $L = 6k$ iterations, after which it is reset to a new randomly initialized listener. At the start, each listener is considered to have already experienced a different number of iterations, uniformly distributed between 0 and $L\frac{N-1}{N}$, inclusive — maintaining a diverse population of listeners with different amounts of experience. This creates a staggered reset regime for a population of listeners.

The speaker's output is given to all the listeners on each round. Each listener guessing the target correctly gets rewarded $r = 1$, otherwise $r = 0$. The speaker on each round gets the mean reward of all the listeners. For evaluation of ease-of-teaching, it is still conducted with a single randomly initialized listener after training for 300k iterations, which might benefit training.

### 5.2 Experiments with Different Population Sizes

We experiment with different listener population sizes $N \in \{1, 2, 10\}$. Note that the sequential reset regime of the previous section is equivalent to the population regime with $N = 1$.

The mean task success rate of the emergent languages for each regime is plotted in Figure 6. We can see that languages are easier-to-teach from the reset regime, then a population regime with 2 listeners, then a population regime with 10 listeners, then the no-reset regime. The average topographic similarity during training is shown in Figure 7. Topographic similarity from the population regime with 2 listeners rises to 0.57, but not as much as the reset regime. For the population regime with 10 listeners, the topographic similarity remains almost at the same level around 0.56.

From the results, we see that although larger populations have more diverse listeners and a less abruptly changing objective, this is not advantageous for the ease-of-teaching and the structuredness of the languages. Moreover, the population regime with a small number of listeners performs closer to the reset regime, while with a relatively large number of listeners seems closer to the no-reset

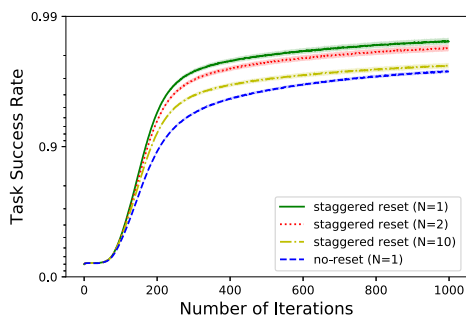
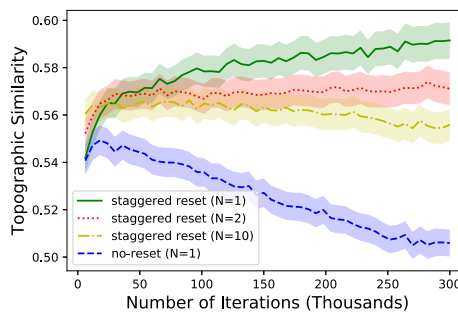

Figure 6: Ease-of-teaching of the languages under staggered reset population regimes.

Figure 7: Comparison of topographic similarity under staggered reset population regimes.

regime. To our surprise, an explicit population of diverse listeners does not seem to create a stronger pressure for the speaker.

## 5.3 Experiments with Simultaneously Resetting All Listeners

The sequential reset regime can be seen as simultaneously resetting all listeners in a population with $N = 1$. We further experiment with resetting all listeners at the same time periodically and a baseline no-reset regime with a population of listeners $N \in \{2, 10\}$.

The mean task success rate and topographic similarity of the language when $N = 2$ are plotted in Figure 8 and Figure 9. The ease-of-teaching curve of simultaneous reset is slightly higher than staggered reset during the first 200 iterations, however staggered reset achieves higher success rates afterwards. The languages from the no-reset regime are less easier-to-teach. While the topographic similarity rises to 0.59 when simultaneously resetting all listeners periodically, it decreases to 0.53 without resets. With a small population of 2 listeners, simultaneous resetting all listeners and without resets show similar impact as $N = 1$.

As for a population of listeners $N = 10$, the mean task success rate and topographic similarity of the languages are plotted in Figure 10 and Figure 11. Languages are easier-to-teach with simultaneous reset, then staggered reset, then no-reset in the population. The simultaneous reset performs better than the staggered reset regime in ease-of-teaching compared to $N = 2$. The topographic similarity rises to 0.59 when simultaneously resetting all listeners, while it shows a similar trend of slight increase in staggered reset and without resets.

Whenever $N \in \{1, 2, 10\}$, languages are easier-to-teach when resetting all listeners periodically than without resets. Moreover, the structuredness of the languages is the highest from resetting all listeners simultaneously. It is interesting to note that when no listener is reset, larger populations tend to produce easier-to-teach and more structured languages, but to a much smaller degree than when resetting listeners.

## 5.4 Discussion

In this section we investigate further the behavior of the different regimes and give some thoughts on what might cause the emergent languages to be easier-to-teach and more compositional.

To get a better view of what the training procedure looks like when resetting a single listener in different sizes of population, we show the training curves of these regimes in Figure 12. In all the regimes, the speaker achieves a communication success rate over 85% with listener(s) within 6k iterations. In the reset regime with $N = 1$, every 6k iterations the listener is reset to a new one, therefore the training success rate drops down to 20%, the chance of randomly guessing 1 target from 5 objects correctly. For a population of 2 listeners, every 3k iterations 1 of the 2 listeners gets reset, which makes the training success rate drops down to around 55%. As for a population of 10 listeners, every 600 iterations 1 new listener is added to the population while others still understand the current

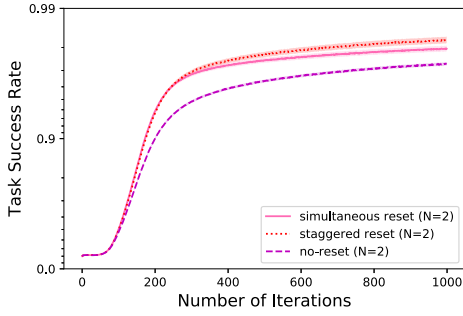

Figure 8: Ease-of-teaching of the languages when the number of listeners $N = 2$.

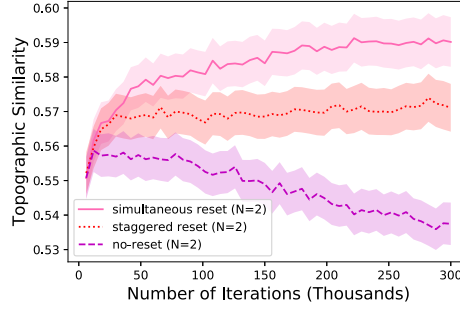

Figure 9: Comparison of topographic similarity when the number of listeners $N = 2$.

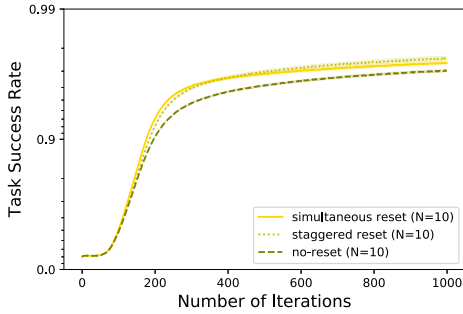

Figure 10: Ease-of-teaching of the languages when the number of listeners $N = 10$.

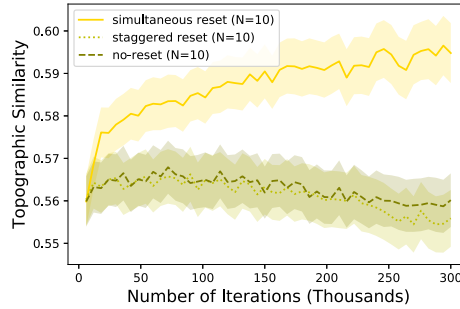

Figure 11: Comparison of topographic similarity when the number of listeners $N = 10$.

language, which makes the success rate only drop to around 82%. For the no-reset regime, the agents maintain a high training success around 90% almost all the time.

When a new listener is introduced, the communication success is lower and the speaker gets less reward and so benefits from increasing entropy in its policy due to the entropy term $H[\pi^S(m|t)]$ in the learning update. The explicitly created pressure for exploration may have the speaker unlearn some of the pre-built communication convention and vary the language to one that is learned more quickly. We plot the speaker's entropy term during training in Figure 13, which backs up this explanation. We can see that there is an abrupt entropy change when a new listener is introduced to a population of 1 or 2 listener(s), which could possibly alter the language to be easier-to-teach. For the population regime with a large number of listeners, we cannot see abrupt changes in the entropy term. Although 1 listener gets reset, the majority of the population maintains communication with the speaker. Thus, the speaker is less likely to alter the communication language to one that the new listener is finding easier to learn.

This explanation also aligns with the result that simultaneously resetting all listeners in different sizes of a population will have better performance. Since simultaneous reset will create a high abrupt entropy instead of being smoothed out by the others in the population when staggering the reset. It would seem that the improvement comes from abrupt changes to the objective rather than smoothly incorporating new listeners. Figures showing entropy changes when simultaneously resetting with a population size of 2 and 10 listeners are in Appendix B.

## 6 Conclusions

We propose new training regimes for the family of referential games to shape the emergent languages to be easier-to-teach and more structured . We first introduce ease-of-teaching as a factor to evaluate emergent languages. We then show the connection between ease-of-teaching and compositionality, and how introducing new listeners periodically can create a pressure to increase both. We further

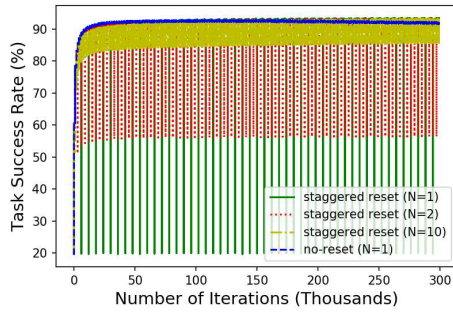

Figure 12: Training curves of the agents in different regimes.

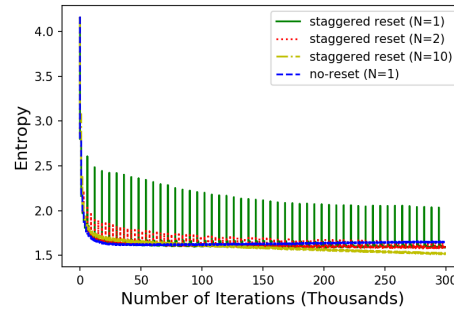

Figure 13: Speaker's entropy in different regimes.

experiment with staggered resetting a single listener and simultaneously resetting all listeners within a population, and find that it is critical that new listeners are introduced abruptly rather than smoothly for the effect to hold.

## Acknowledgments

The authors would like to thank Angeliki Lazaridou, Jakob Foerster, Fei Fang for useful discussions. We would like to thank Yash Satsangi for comments on earlier versions of this paper. We would also like to thank Dale Schuurmans, Vadim Bulitko, and anonymous reviewers for valuable suggestions. This work was supported by NSERC, as well as CIFAR through the Alberta Machine Intelligence Institute (Amii).

## Footnotes

[1]Note that there is no perceptual problem in our experiments.

[2]The entropy term here is a token-level approximation computed only for the sampled symbols except for the last symbol in the message.

[3]Note that communication actions (i.e., speaking and listening) are the only actions in referential games. In such cases, successful task completion ensures "useful communication is actually happening" (Lowe et al., 2019).

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
