[Supplementary Material · eot_appendix.pdf]

# A   Choosing Hyperparameters

Successful communication between agents is often achieved relatively quickly and consistently with hyperparameters $\lambda^S$ and $\lambda^L$ from {0.05, 0.1}. We run experiments of training 1 listener with or without resets using different combinations of values for $\lambda^S$ and $\lambda^L$, and observe the ease-of-teaching of the languages after training. The ease-of-teaching curve of the languages from the reset regime and the no-reset regime when $\lambda^S = 0.05$ and $\lambda^S = 0.1$, are shown respectively in Figure 14 and Figure 15. We find that when $\lambda^S = 0.05$, exploration in the objective is too small to show the impact of bringing new listeners into the training regime, compared to $\lambda^S = 0.1$. Therefore, $\lambda^S = 0.1$ is chosen for our main experiments.

$\lambda^L = 0.1$ is not chosen since in the no-reset regime after training for 220k iterations the success rate drops to 20% occasionally, which gives the no-reset regime a huge variance in performance. $\lambda^L = 0.05$ was more stable for comparing regimes with and without resets.

Figure 14: Ease-of-teaching of the languages when $\lambda^S = 0.05$.

Figure 15: Ease-of-teaching of the languages when $\lambda^S = 0.1$.

# B   Entropy of Population Experiments

Entropy in the population experiments when $N = 2$ and $N = 10$ are plotted in Figure 16 and Figure 17. For simultaneous reset, large spikes are shown. For staggered reset, spikes are visible only when $N = 2$. These observations coincided with the ease-of-teaching and compositionality measure of the languages.

# C   Experiments on a Larger Object Set

We also experiment on a larger set of objects over 100 random seeds, with 3 attributes and 8 values per attribute, resulting in $8^3 = 512$ objects and a message length of 3. Sensibly, we increased the

Figure 16: Speaker's entropy in different regimes when $N = 2$.

Figure 17: Speaker's entropy in different regimes when $N = 10$.

Figure 18: Ease-of-teaching of the emergent languages with and without resets.

Figure 19: Comparison of topographic similarity with and without resets.

Figure 20: Changes of topographic similarity with respect to training task success rate.

iterations between resets to give more time to learn in this more challenging communication task (15k instead of 6k). The same trends continue: see Figure 18 and Figure 19.

## D Changes of Topographic Similarity during Training

The metrics of ease-of-teaching and topographic similarity are both obtained during evaluation. But how does the reset regime affect the training task success rate? Here we show the changes of topographic similarity with respect to training task success rate in Figure 20. Data points are from the training iterations just before we freeze the speaker's parameters. We measured the topographic similarity of deterministic languages the same as before. We can see that the languages from the reset regime achieves higher task success during training as well. Furthermore, for the languages from the staggered reset regime of 1 or 2 listener(s), with training task success increases, topographic similarity increases. While for the no-reset regime, maintaining a similar training task success around 92%, topographic similarity drops from 0.54 to 0.52.

## E Comparison Between Simultaneous Reset for Different Population Sizes

We compare ease-of-teaching and topographic similarity of the simultaneous reset regime for a population of $N \in \{1, 2, 10\}$ listener(s) in Figure 21 and Figure 22. Note that when $N = 1$ the simultaneous reset regime is equivalent to the sequential reset regime in Section 4, and the staggered reset regime in Section 5.2. We find that for simultaneous reset, a population of listeners does not help ease-of-teaching at all, but may help achieve a slightly better topographic similarity.

Figure 21: Ease-of-teaching of the emergent languages under simultaneous reset regimes.

Figure 22: Comparison of topographic similarity under simultaneous reset regimes.

# F   Computing Infrastructure

Our experiments are run on NVidia® GeForce® GTX TITAN X and NVidia® GeForce® GTX 1080 FE (Pascal) graphics cards.