[Reviews · NeurIPS 2019]

Reviewer 1



Overall, the paper was clearly written and had high experimental standards. However, the setting was simple, and it was unclear if the results would apply in more complex language emergence settings. The results about the population setting raise interesting questions that should be further explored. *Originality* Although I think this work is sufficiently different, there has been some past work on iterated learning with artificial agents, references below: - Kirby, 1998: Learning, Bottlenecks and the Evolution of Recursive Syntax - Smith and Hurford, 2003: Language Evolution in Populations: extending the Iterated Learning Model - Vogt, 2005: The emergence of compositional structures in perceptually grounded language games I think the paper should either change the wording in lines 37-40 to not give the impression that iterated learning has only been explored for human languages, or to refer to some of these works if relevant. I do think that this paper is different enough from those works: the listener resetting idea here differs from iterated learning where a listener becomes a speaker, and the agent architectures and communication protocols here follow current neural emergent communication research. *Quality* The experiments are thorough, averaging over a range of random seeds. One shortcoming of the work is that the space of possible inputs and messages is very simple: inputs are purely symbolic, and there are only two attributes, and two tokens in the messages. This has a couple limiting effects, which make it unclear whether the investigation would also apply in language emergence settings that afford more compositional languages or have non-trivial perceptual problems: 1) the language is only investigated for compositionality (and only capable of being compositional) in a very limited sense: there is no significant hierarchical structure, as the metric in effect computes the degree to which each of the two tokens in the message consistently and uniquely identify each of the two attributes. 2) the Levenshtein and cosine similarities used in the topographic similarity calculation can only take on three discrete values, and there are only 32 possible configurations of objects in the world. I'm a bit concerned that a language could obtain a high topographic similarity just by chance. This is mostly addressed by providing the permuted baseline and averaging over a large number of random trials, but I'd be more convinced if some experiments were done to show that the effects are also robust in a setting with more attributes and more tokens passed in the message (to produce a larger possible number of similarity values), or a larger number of values for each attribute in this two attribute setting (which would increase the number of reference--message pairs used to compute the correlations). In addition, I felt that the experiments on population didn't add much to the paper in their current form. While experimenting with populations is well-motivated and an interesting thing to try, the positive result (from simultaneous resetting of the population) seems to produce an effect on a similar scale to just having a single agent that is reset, and the negative results (that staggered resetting of a population doesn't produce compositionality as effectively as using a single listener) could be more thoroughly investigated: - How much does the speaker's language change throughout training, in both the resetting and population settings? - Is the entropy pattern for "reset all of 2" or "reset all of 10" similar to the pattern for "reset"? This comparison didn't seem to be done, but if this is true, it would lend more support to the smoothing hypothesis. *Clarity* The paper was clear overall, with a few places for improvement (see minor comments below). *Significance* See "contributions" above. *Minor comments and clarifications*: - Are the listeners are reset to their original parameters, or to a new random parameter configuration. This wasn't clear to me from the text, and could impact performance, as the speaker was trained for some time against the listener with parameters close to the original configuration. - 19: "have made to coax agents" -> have coaxed agents - 88: is the entropy term \pi^S the entropy of the distribution over full message sequences, or a token-level approximation computed only for the values of m_1 that are sampled (which I understand is commonly used in RL)? The former seems more appropriate for the entropy investigation in 5.4. - 109: I think it would be simpler to present Levenshtein distance as Hamming distance (they should be equivalent since all messages are of length 2). - For the ease-of-teaching experiments on permuted languages in section 3, it seems that the teacher is likely frozen during this evaluation (so that the language is fixed), but this was unclear to me from the text. - The presentation of inputs in terms of colors and shapes in 2.1 seems irrelevant, as the inputs to the agents use one-hot encodings for each color and shape. I'd consider just replacing colors and shapes with symbols (e.g. letters, rather than colors, and numbers, rather than circles) in Table 1 and 2, or if that's not done, the text should make it clearer that there is no perceptual problem here for the agents to solve, and the agents just receive mutually-exclusive symbolic inputs. - I felt Table 1 was more useful as an illustration of topographic similarity, than as a demonstration of the success of the reset regime. This language has substantially higher topographic similarity than the mean produced by resetting, and it seems that high topographic similarity, as it's defined for this task, will necessarily result in a symbol to attribute mapping like the one seen here. The description is also not totally correct: for shapes, there are exceptions to disjointness, e.g. "a" is as likely to reprsent "square" as it is "triangle". - It would be interesting to have some analysis of how much the language changes in the reset regime between listeners. - 180: "affect" -> "effect" - The experimental setup in 5 was a bit unclear: my assumption is that the "reset 1 of X" experiments in Figures 6-11 correspond to the staggered resetting of *all* agents described in 5.1, based on the discussion in 5.4, but I that this should be made clearer earlier in the text. I'd also suggest a different name than "reset 1 of X" (such as "staggered resetting") -- the current phrasing suggests that only a single listener is ever being reset and the others persist through all 300k iterations. Some details about the ease-of-teaching evaluation in 5.2 were also unclear: - Does the evaluation follow 4.1, i.e. is it evaluating the speed of learning of a single, randomly-initialized agent? - Does the evaluation in 5.2 happen after 300k iterations? - It would help to make the same clarification in the Figure 8 caption that is also made in line 212. - I think it would be useful to have some investigation of how the reset-all population experiments parallel the reset a single agent experiments, as it seems these have broadly similar topographic similarity results, and might be indicative of some averaging effect. --- Update after author response --- Thank you for the response and the clarifications, and for these additional experiments and analysis. The experiments with more referents help to address my concern about complexity, and the investigation of the entropy term in the multi-agent case helps explain similarities between the single-agent and non-staggered setting. I've raised my score from a 5 to a 6.

Reviewer 2



Originality: The main insight (compositional languages are easier to teach) is a creative way to draw from evolution-of-language literature. The results that ease-of-teaching encourages compositionality in artificial emergent language are quite interesting from a scientific perspective, and it doesn't seem to have been explored previously. Significance: The main insight (compositional languages are easier to teach) is interesting and well validated. I would've been excited to see the work take advantage of this insight/result more, as it seems to open a number of promising avenues. For instance, what about training the speaker using RL or meta-learning on the "ease-of-teaching" metric itself (i.e., accuracy of a new listener after a few updates)? Or what about using topological similarity as an auxiliary reward to increase the speed of teaching? Quality: The work validated it's mainly hypothesis thoroughly. I felt that Figs. 3-5 captured the main results pretty effectively, and that other figures / tables could have been consolidated and/or moved to the appendix, as they seem to make similar points (which I was already convinced of). This would've left more room for exploring how to use "ease-of-teaching" more effectively or as an explicit objective (as I discussed a bit earlier) - using "ease-of-teaching" more aggressively, it seems possible to get impressive empirical resutls (i.e., the longest compositional sentences in emergent communication thus far). Clarity: The paper is easy to follow. The authors make intuitive design decisions, lay out clear explanations of the results, including counter-intuitive ones. I think the paper can be compressed: e.g., Tables 1 and 2 are their descriptions can be moved to the Appendix and Figs. 6/8/10 can perhaps be consolidated (as well as Figs. 7/9/11). After Author Response: I have read the other reviews and the author response, and I stand by my evaluation of this paper. The rebuttal mainly addressed R2's concerns. Increasing my score would require using/optimizing "ease of teaching" in an additional, substantial way (i.e., directly optimizing ease-of-teaching), which likely requires a good amount of additional experimentation.

Reviewer 3



** Summary The paper takes the speaker-listener population setup to study the emergence of language structure in the form of compositionality and ease of teaching. This is done via constant replacement of listeners that creates a pressure for solutions to tilt that way. Interestingly, smooth population changes were not as amenable for such emergence when compared to abrupt changes. ** Strength - The paper performs an extensive set of experiments. The population experiments were quite interesting. - Experimental setup is well described and the attached source code was quite useful ** Weakness - Not completely sure about the meaning of the results of certain experiments and the paper refuses to hypothesize any explanations. Other results show very little difference between the alternatives and unclear whether they are significant. - Lot of result description is needlessly convoluted e.g. "less likely to produce less easier to teach and less structured languages when no listener gets reset". ** Suggestions - A related idea of speaker-listener communication from a teachability perspective was studied in [1] - In light of [2], it's pertinent that we check that useful communication is actually happening. The differences in figures seem too small. Although the topography plots do seem to indicate something reasonable going on. [1]: https://arxiv.org/abs/1806.06464 [2]: https://arxiv.org/abs/1903.05168

[Author Response · NeurIPS 2019]



Figure 1: Ease-of-teaching.

Figure 2: Topographic similarity.

Figure 3: Speaker's entropy ($N = 2$).

First, we want to thank all the reviewers for the helpful comments and suggestions which will be reflected in future
iterations.

**R2**: We have since ran further experiments with 3 attributes and 8 values per attribute, resulting in $8^3 = 512$ objects
and a message length of 3. Sensibly, we increased the iterations between resets to give more time to learn in this more
challenging communication task (15k instead of 6k). The same trends continue: see Figure 1 and Figure 2.

The primary contribution of the population experiments is to explore whether the observed improvements come from
the sequence of agents acting as a population. We found instead it is rather the abrupt change in listener that matters for
the effect. The pattern of the entropy term for non-staggered resets does match the single-agent reset case (see Figure 3
for the $N = 2$ case). We will make both of these observations more clear and include such graphs to strengthen this
hypothesis as suggested.

Related works in iterated learning should be discussed more thoroughly. We will change the wording and include a
discussion of the papers you noted.

An initial investigation into the degree of language change suggests that the amount of change in the language during
training is similar across all regimes (with and without reset), with as much as half the language changing in 6k
iterations.

We had a transcription error in Table 1 in the paper so the whole third column of the table should read "be ee ce ge".
The language shown is to compare the most structured languages we could find after training 300k iterations, with
ones from the no-reset regime.

Finally, we want to confirm that your other assumptions in "minor comments and clarifications" are correct and
will endeavour to clarify these points in the text: all the listeners are reset to random parameters; we calculated the
entropy term with a token-level approximation; for ease-of-teaching evaluation the teacher is frozen; in the population
experiments ease-of-teaching is evaluated after 300k iterations; "reset 1" needs to be clarified and we will use your
"staggered" nomenclature; the results of "simultaneous resetting" in $N = 1, 2, 10$ are all similar and statistically
inseparable.

**R3**: We should be clear that the reviewer's third described contribution — the metric of topographic similarity — is
not novel to this work. On the other hand, the metric of ease-of-teaching is a new metric of evaluation and we think it
is a valuable contribution to the emergent communication field.

Thanks for pointing out several avenues this work opens. We also think explicitly optimizing ease-of-teaching is a
worthwhile direction of future work.

**R4**: We will add a brief discussion of the papers you mentioned. Note that communication actions (i.e., speaking
and listening) are the only actions in referential games. In such cases, successful task completion ensures "useful
communication is actually happening".

[Meta-Review · NeurIPS 2019]

The reviewers appreciated the rigor and clarity of the paper. In their reviews, they have listed several references that are worth to be included and discussed in the final version of the paper.